# Extracellular Vesicles from Human Umbilical Cord-Derived MSCs Affect Vessel Formation In Vitro and Promote VEGFR2-Mediated Cell Survival

**DOI:** 10.3390/cells11233750

**Published:** 2022-11-24

**Authors:** Ana Muñiz-García, Bettina Wilm, Patricia Murray, Michael J. Cross

**Affiliations:** 1Department of Molecular Physiology and Cell Signalling, Institute of Systems, Molecular and Integrative Biology, University of Liverpool, Liverpool L69 3BX, UK; 2Department of Pharmacology and Therapeutics, Institute of Systems, Molecular and Integrative Biology, University of Liverpool, Liverpool L69 3GE, UK

**Keywords:** extracellular vesicles, MSC, VEGFR2, AKT, apoptosis, angiogenesis

## Abstract

Mesenchymal stromal cell (MSC)-derived extracellular vesicles (EVs) have emerged as novel tools in regenerative medicine. Angiogenesis modulation is widely studied for the treatment of ischaemic diseases, wound healing, and tissue regeneration. Here, we have shown that EVs from human umbilical cord-derived MSCs can affect VEGFR2 signalling, a master regulator of angiogenesis homeostasis, via altering the phosphorylation of AKT. This translates into an inhibition of apoptosis, promoting exclusively cell survival, but not proliferation, in human microvascular endothelial cells. Interestingly, when comparing EVs from normoxic cells to those obtained from hypoxia (1% O_2_) preconditioned cells, hypoxia-derived EVs appear to have a slightly enhanced effect. Furthermore, when studied in a longer term endothelial-fibroblast co-culture angiogenesis model in vitro, both EV populations demonstrated a positive effect on vessel formation, evidenced by increased vessel networks with tubes of significantly larger diameters. Our data reveals that EVs selectively target components of the angiogenic pathway, promoting VEGFR2-mediated cell survival via enhancement of AKT activation. Our data show that EVs are able to enhance specific components of the VEGF signalling pathway and may have therapeutic potential to support endothelial cell survival.

## 1. Introduction

Extracellular vesicles (EVs) are membranous nanoparticles naturally released by most, if not all, cell types. EVs can be classed into two main subtypes according to their biogenesis: exosomes and ectosomes [1,2,3]. Exosomes are generated through the endocytic pathway, having their origin as intraluminal vesicles (ILVs) within the multivesicular endosomes (MVEs); these ILVs are termed exosomes once they are released to the extracellular space following MVE fusion with the plasma membrane. As a consequence of their biogenesis process, exosomes are usually the smallest of the EVs, ranging from 30 to 150–200 nm [2,4,5]. On the other hand, ectosomes are a more heterogenous population comprising EVs generated through plasma membrane budding, therefore including microvesicles and oncosomes [2,3]. Ectosomes have varying sizes (from 50 nm to 10 µm), sometimes overlapping with exosomes, thus making it quite difficult at times to differentiate between both subpopulations without investigating their origin. For this reason, in the present study the nomenclature chosen to describe the EV population under investigation was based on their size, assuming a mixed population of those EVs in the lower size range (small EVs).

In the last decade, EV-related studies have significantly increased due to their demonstrated role in cell-to-cell communication [6,7]. The most remarkable characteristic of EVs is their capacity to carry all kinds of active biomolecules (proteins, lipids, and nucleic acids) allowing autocrine and paracrine signalling in cells [2,7]. Therefore, they are not only a source of information regarding cellular physiology, but also offer great potential in different biomedical scenarios, including their use as biomarkers, therapeutics, or drug carriers [8,9,10]. Interestingly, EV cargo has been suggested to be influenced by both the cell type of origin and the environmental or cell-state conditions [6], therefore conditioning the impact EVs would have in the recipient cells. For this reason, EVs are of great interest in the regenerative medicine field as a potential alternative to cell therapies.

Mesenchymal stromal cells (MSCs) have been widely studied as potential therapeutic tools. MSCs were initially discovered in the bone marrow, but are also present in other tissues such as the umbilical cord and adipose tissue [11]. MSCs have demonstrated promising results as potential therapeutics, showing both regenerative and immunomodulatory capacity [11]. However, their mechanisms of action and long-term safety remain to be fully elucidated [12]. Indeed, the following properties of MSCs have been reported: potential tumorigenesis [13]; differentiation towards undesired cells [14]; reduced engraftment [15], potentially caused by their entrapment in the lung after intravenous injection [15,16]; or immunogenicity, despite being considered immune-privileged cells [17].

Mounting evidence suggesting that the cellular source highly impacts on EV properties soon led to the hypothesis that MSC-derived EVs maintain the therapeutic value of the cells of origin, or that they are even the ultimate mediators of their effects in vivo [18]. Since EVs are acellular bioactive particles, they bypass some of the limitations associated with cell therapies. Consequently, EVs have become of great interest in the field of regenerative medicine and have been reported to promote the repair of cardiac, liver, muscle, kidney and nervous tissue in vitro and in animal models [19,20]. Despite the growing investigations on MSC-derived EVs and their therapeutic benefit, the mechanism of action of these nanometric particles still remains to be fully clarified.

Among the properties attributed to MSCs and MSC-derived EVs is the stimulation of angiogenesis [21]. Angiogenesis modulation is of particular interest to promote wound-healing or tissue regeneration. Indeed, MSCs and MSC-derived EV therapies have been suggested as potential treatments for injuries to the vasculature, such as occurs in kidney injury, stroke, or cardiac infarction [22]. However, the mechanisms whereby MSC-derived EVs promote angiogenesis are still unknown. One of the central molecular pathways involved in sprouting angiogenesis is the vascular endothelial growth factor (VEGF) signalling cascade [23]. This process is mainly mediated by VEGF receptor 2 (VEGFR2) that, upon binding with VEGF, will auto-phosphorylate, prompting the initiation of a sequence of signalling cascades responsible for cell proliferation, migration, and survival [23,24].

Here, we investigated the angiogenic potency of EVs obtained from human umbilical cord-derived mesenchymal stromal cells (hUCMSCs) and their potential mechanism of action via VEGF signalling. Furthermore, considering that hypoxia is one of the most potent promoters of angiogenesis and we have previously shown that preconditioning of hUCMSCs to hypoxia affects EV release [25], we also investigated whether hypoxia would enhance the pro-angiogenic effect of their derived EVs.

## 2. Materials and Methods

### 2.1. Antibodies and Reagents

Primary antibodies against VE-cadherin (#2158), phospho-VEGFR2 at Tyr1175 (D5B11, #3770), VEGFR2 (55B11, #2479), phospho-AKT at Ser473 (D9E, #4060), AKT (C67E7, #4691), phospho-ERK1/2 (p44/42 MAPK, 137F5, #4695), GAPDH (D16H11, #5174) and pan-Actin (D18C11, #8456) were from Cell Signalling Technology (CST; Danvers, MA, USA). The antibody anti-RCAN1 (#D6694) was from Sigma-Aldrich (St. Louis, MO, USA), anti-CD81 (MCA1847) was from BioRad (Hercules, CA, USA), anti-CD9 (#10626D) was from ThermoFisher (Waltham, MA, USA), anti-CD63 (#556019) was from BDBiosciences (Franklin Lakes, NJ, USA), and anti-GRP78 was from Serotec (Raleigh, NC, USA). Secondary anti-rabbit or anti-mouse Horseradish Peroxidase (HRP)-coupled antibodies were from Jackson Immunoresearch (West Grove, PA, USA) and anti-rabbit Alexa488 antibody (#A-21206) was from Invitrogen (Waltham, MA, USA).

Human recombinant VEGFA_165_ was obtained from Proteintech (#HZ-1038; Rosemont, IL, USA).

### 2.2. Cell Culture

Human umbilical cord-derived mesenchymal stromal cells (hUCMSCs) were obtained from NHS Blood and Transplant (NHSBT) at passage 2 and were amplified in minimum essential medium α (MEMα) containing GlutaMAX (32561-029; Gibco, Waltham, MA, USA) supplemented with 10% foetal bovine serum (FBS; 10270-106, Gibco). hUCMSCs were used up to a maximum of passage 7. Prior to EV isolation studies, cells were serum-starved in MEMα medium and either maintained in a regular humidified air incubator (approx. 90–95% humidity) set at 5% CO_2_ and 37 °C (normoxic conditions, NOR) or transferred to a H35 hypoxia workstation (Don Whitley Scientific, Bingley, UK) at 1% O_2_ *v*/*v*, 5% CO_2_ *v*/*v*, 94% N_2_ *v*/*v* and 75% humidity at 37 °C (hypoxic conditions, HYP).

Normal human dermal fibroblasts (NHDF) and human dermal microvascular endothelial cells (HDMEC), were obtained from Promocell (C-12300 -juvenile-, and C-12210 -juvenile-, respectively). NHDFs were grown in Fibroblast Growth Medium 3 (FGM3) consisting of Fibroblast Basal Medium 3 supplemented with 10% foetal calf serum (FCS), 1 ng/mL recombinant human basic Fibroblast Growth Factor (bFGF), and 5 μg/mL recombinant human insulin (C-23130, Promocell, Heidelberg, Germany). HDMECs were amplified in Endothelial Cell Growth Medium MV2 (ECGM-MV2) containing Endothelial Cell Basal Medium MV2 (ECBM-MV2) supplemented with 5% FCS, 5 ng/mL recombinant Human Epidermal Growth Factor, 10 ng/mL bFGF, 20 ng/mL Long R3 Insulin-like Growth Factor, 0.5 ng/mL recombinant human Vascular Endothelial Growth Factor A 165 (VEGFA_165_), 1 μg/mL ascorbic acid, and 0.2 μg/mL hydrocortisone (C-22121, Promocell, Heidelberg, Germany). Cells were routinely cultured on 0.5% (*w*/*v*) gelatin-coated plates (G1890, Sigma-Aldrich, St. Louis, MO, USA) in normoxic conditions.

### 2.3. Extracellular Vesicles Isolation and Analysis

Following 24 h serum-starvation of hUCMSCs under normoxic or hypoxic conditions, extracellular vesicles (EVs) were isolated from conditioned medium using differential ultracentrifugation as previously described [25,26]. Briefly, cell debris were discarded by centrifuging at 2000× *g* for 10 min; larger vesicles were separated with a 10,000× *g* centrifugation for 30 min; and smaller EVs were obtained after two consecutive 100,000× *g* centrifugations of 75 to 120 min, performing a PBS wash in between both centrifugations. The final EV pellet was resuspended in PBS and either analysed or stored at −80 °C in small aliquots until use. EVs were quantified using nanoparticle tracking analysis (NTA) with a Nanosight NS300 instrument (Malvern Panalytical Ltd, Malvern, UK) following 3 captions of 60 s each. EV isolation was further validated using transmission electron microscopy (TEM) and western immunoblotting (WB).

#### 2.3.1. Transmission Electron Microscopy (TEM)

10 µL of EVs in suspension were placed onto a copper carbon coated glow discharged TEM grid and incubated at room temperature for 10 min. Grids were washed over two 100 µL drops of PBS before fixation with 1% glutaraldehyde in PBS for 5 min. Next, grids were washed over 2 drops of ddH_2_0 before staining with 1% aqueous uranyl acetate (UA) for 30 s. Excess UA was wicked away with filter paper and grids were dried. The prepared grids were then viewed at 120 KV on a FEI Tecnai G2 Spirit (FEI, Hillsboro, OR, USA) with Gatan RIO16 digital camera (Gatan Inc., Milton, UK).

#### 2.3.2. Western Immunoblotting (WB) for EV Analysis

EVs were characterised via WB by preparing samples containing approximately 2 × 10^9^ EVs (as per NTA counting). In parallel, these were compared to samples containing 20 µg and 5 µg of whole cell lysate from the hUCMSCs from which EVs were derived. All samples were diluted in non-reducing loading buffer (LDS Sample Buffer (NP0008, Invitrogen) without β-mercaptoethanol) and boiled at 90 °C for 5 min. From this step on, samples were immunoblotted as later described in Section 2.5.

### 2.4. Acute Stimulation Studies

For acute stimulation experiments, HDMECs were grown in 12-well plates until fully confluent and serum-starved in ECBM-MV2 supplemented with 1% (*v*/*v*) FCS for at least 16 h. Following the starvation period, cells were pre-treated for 30 min or 24 h with EVs at 2 × 10^7^ EVs/well, and followed by 50 ng/mL dose of VEGFA_165_ for different periods of time.

### 2.5. Cellular Lysates and Western Immunoblotting

Cellular lysates were obtained at the end of each experiment using ice-cold freshly made RIPA buffer (45 mM Tris pH 7.5, 135 mM NaCl, 45 mM NaF, 1.8 mM EDTA, 9% (*w*/*v*) glycerol, 1% (*w*/*v*) Triton X-100, 1 mM Na_3_VO_4_, and 1% (*v*/*v*) protease inhibitor cocktail (P8340, Sigma)) by keeping cells in agitation for 15 min for a complete lysis, scraping the cells, and centrifuging at 17,000× *g* for 20 min (all steps performed on ice or at 4 °C). The obtained supernatant was used to prepare the samples for loading in LDS Sample Buffer (NP0008, Invitrogen) containing 2.5% (*v*/*v*) β-mercaptoethanol. Each sample was boiled at 90 °C for 5 min, resolved in Tris-Glycine gels, transferred to nitrocellulose membranes (Hybond C, GE Healthcare, Chicago, IL, USA), and blocked with 5% (*w*/*v*) Bovine Serum Albumin Fraction V (BSA; 10735108001, Merck, Rahway, NJ, USA) in Tris-buffered saline (pH 7.6). Blots were probed with primary antibodies and later HRP-coupled secondary antibodies, both diluted in 2% (*w*/*v*) BSA. Chemiluminescence was detected following incubation with Pierce™ ECL Western Blotting (32106, Thermo Fisher, Waltham, MA, USA) in photographic paper (Fuji Medical X-ray Film, Super RX, 100NF; Jet X-ray, London, UK). The densitometry analysis of the obtained bands was performed following digitalisation using ImageJ J (version 1.53k) software (National Institute of Health (NIH), Bethesda, MD, USA).

### 2.6. 3D Collagen Gel Tube Formation Assay

To assess the impact of EV treatments in tube formation, a 3D collagen gel tube formation assay was used. Gels containing PureCol™ type I bovine collagen (#5005-100ML; CELLINK, Boston, MA, USA) were prepared by mixing 10× Ham’s F-12 medium, 0.1 M NaOH, collagen type 1 (ratio 1:1:8), supplemented with bicarbonate solution (Invitrogen) to 0.117% v/v and Glutamax-I (Invitrogen) to 1% *v*/*v*. The gels were allowed to set at 37 °C overnight in the incubator. Serum-starved HDMECs were seeded at 90,000 cells per well in 24-well plates in the presence or absence of 10^7^ EVs/well prior to the addition of the top layer of collagen. The top gel was allowed to set for two hours prior to cells being further stimulated with 50 ng/mL VEGFA_165_ and/or 10^7^ EVs/well for 24 h. Treatments were added over the top collagen layer diluted in ECBM-MV2 1% FCS but at 2× concentration to counter collagen diffusion. Tubular length was quantified using the *Angiogenesis Analyzer* plugin [27] for Image J (version 1.53k) software (NIH, Bethesda, MD, USA).

### 2.7. Cell Proliferation Assay

Impact of EVs on cellular proliferation was quantified using a luminescence assay. First, HDMECs were seeded at 12,000 cells per well in gelatin-coated 24-well plates with full growth medium. Next day, cells were serum starved on ECBM-MV2 supplemented with 1% (*v*/*v*) FCS overnight. Following starvation, cells were pre-treated with 2 × 10^7^ EVs/mL for 30 min and further stimulated with VEGFA_165_ (50 ng/mL) or FCS (10%) for a period of 3 days. Cells were then washed in PBS with Ca^2+^/Mg^2+^ prior to the addition of CellTiter-Glo™ reagent (#G7571; Promega, Madison, WI, USA) following the manufacturer’s instructions. Luminescence was measured on white-walled flat-bottomed 96-well plates (Greiner) using a plate reader.

### 2.8. Apoptosis Assay

Relative activity of the executioner caspases, caspase-3/7, was detected in HDMECs undergoing tubular morphogenesis within collagen gels (described in Section 2.6) using a luminescent assay. Each condition tested was prepared in triplicate wells. Following 6 h treatment ± VEGFA and ± EVs, media and the top layer of collagen was removed from the plates by direct aspiration. Cells were then washed in PBS with Ca^2+^/Mg^2+^ prior to the addition of the Caspase-Glo^®^ 3/7 reagent (#G8091, Promega, Madison, WI, USA) following the manufacturer’s instructions. Cells were incubated at room temperature with orbital shaking for 1 h. Luminescence was measured in duplicate from each condition on white-walled flat-bottomed 96-well plates (Greiner) using a plate reader.

### 2.9. Endothelial-Fibroblast Co-Culture Angiogenesis Assay

NHDFs were seeded in FGM3 at 20,000 cells/well on gelatin-coated coverslips at the bottom of 24-well plates and incubated for 3 days. On day 4, HDMECs were added on top of the NHDF monolayer at 20,000–30,000 cells/well in ECGM-MV2 medium. On day 5, cells were serum-starved with ECBM MV2 containing 1% (*v*/*v*) FCS treated with 10^7^ EVs/well and/or 50 ng/mL of VEGFA_165_. These treatments were added again freshly on day 8, to finally terminate the experiment on day 10. By the end of the experiment, cells were subsequently fixed in 2% paraformaldehyde/PBS for 20 min, washed with PBS, permeabilised with 1% Triton X-100/PBS for 15 min and blocked with 1% (*w*/*v*) BSA, 5% (*v*/*v*) donkey serum (D9663, Sigma), 0.1% Tween-20 in TBS for 30 min. Cells were immunostained with anti-VE-cadherin primary antibody diluted 1:400 in 1% (*w*/*v*) BSA, 0.1% Tween-20 in TBS for 1 h 30 min under shaking conditions at room temperature, followed by an incubation with anti-rabbit Alexa488 secondary antibody diluted 1:1000 in 1% (*w*/*v*) BSA, 0.1% Tween-20 in TBS for 1 h 30 min under mild shaking conditions, at room temperature and covered from the light. Coverslips were mounted on microscope slides using ProLong™ Gold Antifade Mountant with DAPI (P36935, Invitrogen, Waltham, MA, USA), sealed, and stored at 4 °C until imaging. Images were obtained using a Zeiss AxioObserver Z1 (Zeiss, Jena, Germany) epifluorescent inverted microscope with Apotome2 and acquired using ZenPro 3.3 software (Zeiss, Jena, Germany). A number of fields were chosen at random for each condition and imaged using Plan-Neofluar 5X/0.15, or Plan-Apochromat 20X/0.8 objectives. Images were quantified using open-source software AngioTool (version 64 0.6a) [28] and REAVER [29].

### 2.10. Statistical Analysis

Statistical analysis was performed using GraphPad Prism 9 software (Dotmatics, San Diego, CA, USA). Each data set was analysed with the most appropriate test and is indicated in each figure legend. In all cases, the confidence interval was set at 95% and statistical significance was set at *p* < 0.05.

## 3. Results

### 3.1. Characterisation of Small Extracellular Vesicles from hUCMSCs

In order to study the regenerative properties of extracellular vesicles (EVs) from hUCMSCs, EVs were isolated from the conditioned medium of cells in culture following 24 h of serum starvation under normoxic (NOR) or hypoxic (1% O_2_; HYP) environmental conditions. EVs were isolated via differential ultracentrifugation (dUC), to enrich for smaller EVs. EV characterisation was performed by (i) nanoparticle tracking analysis (NTA), (ii) transmission electron microscopy (TEM), and (iii) Western blot (WB). Both NTA and TEM data confirmed that the EV population size was within the range of that expected for smaller EVs, and that hypoxic preconditioning did not significantly affect EV size distribution (Figure 1a,b). EV enrichment was also confirmed by WB, with EVs showing enrichment of the 3 classically EV-associated tetraspanins: CD63, CD81 and CD9, but not other cellular and organelle specific markers such as the cytoskeletal protein actin, and the endoplasmic reticulum (ER)-associated chaperone GRP78 (Figure 1c).

### 3.2. hUCMSC-Derived EVs Affect VEGFR2-Mediated Phosphorylation of AKT

MSC-EVs from different sources have been reported to enhance angiogenesis (reviewed in [30]). To assess whether the hUCMSC-derived EVs have any impact on endothelial cell signalling, we analysed the effect of the EVs on a number of VEGF-stimulated intracellular signalling pathways which regulate endothelial cell physiology [23,24]. VEGF-mediated activation of VEGFR2 was assessed by analysing phosphorylation of the C-terminal Tyr 1175 residue, and by detecting the downstream phosphorylation of AKT and ERK1/2. We also analysed the induction of RCAN1.4 which has been shown to be induced in a VEGFR2-specific manner in endothelial cells [31,32]. Stimulation with VEGF induced a rapid phosphorylation of VEGFR2, which progressively declined parallel to the time-dependent decay of total VEGFR2 levels, due to the receptor internalisation and degradation, as previously described [32,33]. Likewise, AKT and ERK1/2 phosphorylation occurred transiently concomitant to VEGFR2 activation (Figure 2). In order to assess both more immediate (i.e., receptor driven responses) and slower (i.e., transcriptional changes) signalling effects, EV pre-treatments were performed for 30 min or 24 h. The shorter EV pre-treatments (30 min) resulted in an alteration in the progression of VEGFR2 signalling cascade, with AKT showing significantly increased and prolonged phosphorylation in response to VEGF (Figure 2b). The longer EV pre-treatments (24 h) caused a significant increase in VEGFR2 phosphorylation and increased AKT phosphorylation in a similar pattern to that observed with 30 min pre-treatments (Figure 2c). Interestingly, regardless of the EV pre-treatment length, no changes were observed in ERK1/2 phosphorylation and RCAN1.4 expression, therefore suggesting that both EV populations appeared to increase the AKT branch of VEGFR2 signalling.

### 3.3. hUCMSC-Derived EVs Affect VEGFA-Mediated Tubular Morphogenesis but Not Cell Proliferation

Proliferation and differentiation of endothelial cells constitute distinct processes in the coordination of angiogenesis. We used an in vitro system with HDMECs plated on a gelatin matrix to facilitate proliferation and with a collagen matrix to facilitate tubular morphogenesis in response to VEGF. In order to assess basal, VEGF and serum mediated proliferation, EV treatments were tested on their own and in combination with VEGFA_165_ or foetal calf serum (FCS). HDMECs plated in a gelatin matrix showed a 30% increase in cell number following stimulation with VEGFA_165_, and a two-fold increase when stimulated with FCS. However, EV treatments had no significant effects on either basal, VEGF or serum-induced cellular proliferation (Figure 3a). HDMECs plated in a 3D collagen matrix were allowed to form vessel networks for 24 h in the presence or absence of NOR/HYP EVs and/or VEGFA_165_. Interestingly, EV treated cells showed a small increase in tubular formation when administered on their own; however, this effect was significantly increased when in combination with VEGFA_165_ (Figure 3b). Therefore, these data reveal that both EVs from NOR and HYP hUCMSCs are able to regulate tubular morphogenesis without affecting cell proliferation.

### 3.4. hUCMSC-Derived EVs Suppress Apoptosis during Tubular Morphogenesis

Endothelial cell survival and suppression of apoptosis are critical for efficient angiogenesis [34]. One of the key mechanisms protecting endothelial cells from apoptosis is VEGF-induced activation of AKT signalling [35,36]. Our finding that EVs were able to enhance VEGFR2-mediated AKT phosphorylation suggests they may contribute to the enhanced tubular morphogenesis observed. Activation of the protease family of caspases represents one of the terminal stages of signal transduction pathways leading to endothelial cell apoptosis [37]. In order to confirm whether EVs impact apoptosis suppression, we studied caspase-3/7 activity using a luminescent-based assay and confirmed caspase 3 cleavage by WB in cells undergoing tubular morphogenesis in a collagen gel. Additionally, cells treated with staurosporine (STP), a well-known pro-apoptotic substance [38], were used as a positive control. Addition of EVs to endothelial cells undergoing tubular morphogenesis appeared to reduce caspase3/7 activity under both basal and VEGFA-stimulated conditions and reduced cleavage of caspase 3 by Western blotting (Figure 4). Overall, these data suggest that hUCMSC-derived EVs regulate tubular morphogenesis by promoting endothelial cell survival.

### 3.5. hUCMSC-Derived EVs Promote Vessel Formation and Impact Vessel Diameter in a Long-Term Angiogenesis Model In Vitro

VEGF signalling is known to promote cell proliferation, migration and survival. The previous experiments showed that hUCMSC-derived EVs could promote VEGF-mediated tubular morphogenesis over 24 h and suppressed apoptosis in a collagen gel. In order to determine the effect of the hUMSC-derived EVs on longer-term angiogenesis we utilised an organotypic fibroblast-endothelial co-culture system where HDMECs are plated on a confluent monolayer of human dermal fibroblasts (NHDFs; Figure 5a) [39,40]. Vascularisation was detected by staining for the endothelial-specific marker VE-cadherin and visualised by immunofluorescence. Stimulation of HDMECs with VEGFA_165_ over 5 days induced an increase in endothelial vessel area (Figure 5b,d). Incubation with hUCMSC-derived EVs increased the basal level of vascularisation but did not augment the VEGF-mediated increase in vascularisation. Interestingly, analysis of vessel diameter revealed that hUCMSC-derived EVs increased vessel diameter under basal conditions with no apparent effect when VEGFA was added (Figure 5c,e).

## 4. Discussion

In the last few decades, mesenchymal stromal cell-derived extracellular vesicles (MSC-EVs) have become prospective alternatives to cell therapies in regenerative medicine [41]. Their mechanism of action and the full extent of their biological or therapeutic properties still remain to be fully elucidated. In the current study, we aimed to shed light on the molecular mechanism behind the effect of hUCMSC-derived EVs on angiogenesis. hUCMSCs are of particular interest, not only because of their readily availability, but also because the newborn status of the donors means the phenotype of the cells is less likely to be affected by their lifestyle or the presence of comorbidities. Our approach involved an initial analysis of the effect of EVs on VEGFR2 signalling in microvascular endothelial cells, followed by assessing their effect on proliferation, apoptosis and angiogenesis in established in vitro assays. Our data suggest that hUCMSC-derived EVs promote VEGF-mediated angiogenesis by promoting cell survival. Furthermore, an enhancement of the pro-angiogenic effect was observed following hypoxia pre-conditioning of hUCMSCs.

VEGF plays a critical role in endothelial cell biology and activation of VEGFR2 can regulate multiple signalling pathways in endothelial cells [42,43]. The hUCMSC-derived EVs appear to cause a small transient increase in agonist-stimulated phosphorylation of VEGFR2, which is more pronounced on 24 h pre-incubations with EVs. There does not appear to be a concomitant increase in VEGFR2 levels, suggesting that the EVs may directly affect VEGFR2 phosphorylation. We have previously shown that VEGFR2 phosphorylation can be regulated by the induction of RCAN1.4, which binds to VEGFR2 and facilitates cell surface internalisation of VEGFR2 [40]. However, pre-incubation with EVs did not affect RCAN1.4 induction, precluding this pathway in causing the phosphorylation of VEGFR2. It is possible that EVs may affect VEGFR2 tyrosine kinase activity, enhancing phosphorylation, or perturb the effect of a phosphatase, indirectly enhancing receptor phosphorylation [44]. The enhancement of VEGF-mediated AKT phosphorylation with EVs was significant following acute pre-incubation with EVs and was also evident with the longer-term pre-incubation with EVs.

AKT is activated by phospholipid binding and activation loop phosphorylation at Thr308 by the Phosphoinositide-dependent kinase-1 (PDK1), which is dependent on membrane Phosphatidylinositol (3,4,5)-trisphosphate (PtdIns (3,4,5)P_3_) [45] and by phosphorylation within the carboxy terminus at Ser473 by the mammalian target of rapamycin complex 2 (mTORC2) [46,47]. EVs have been postulated to enhance AKT phosphorylation by allowing shuttling of micro RNAs such as miR-205 that negatively regulates PTEN expression leading to sustained levels of PtsIns (3,4,5)P_3_ and activation of AKT/mTOR pathway [48]. The acute effect of EVs on VEGF-stimulated AKT phosphorylation observed in the HDMECs would suggest that the effect we observed is independent of transcriptional and translational processes regulated by miRNAs and more likely an effect on a direct activator of AKT. Indeed, MSC-derived EVs have been shown to stimulate AKT phosphorylation in human fibroblasts and ketatinocytes independently of miR-205 activity [49].

The most significant effect of EVs on endothelial cell physiology was evident on the collagen gel matrix, where endothelial cells undergo tubular morphogenesis in the presence of VEGFA_165_ (Figure 3). On a collagen gel matrix, endothelial cells stop proliferating and accrue in G0/G1 [40,44]. Activation of AKT is critical in facilitating VEGF-mediated endothelial cell survival when endothelial cells are placed in low-serum, a condition that will cause cell cycle arrest [36]. Taken together, this suggests that EVs may stimulate and also enhance VEGF-mediated tubular morphogenesis by enhancing AKT-mediated cytoprotection and suppression of apoptosis in the endothelial cells in a collagen gel. The ability of EVs to target the AKT pathway has also been reported from in vivo studies: EVs from mesenchymal stromal cells can enhance myocardial viability after reperfusion injury [50]. It has been shown that injection of purified EVs prior to reperfusion increased the level of phosphorylated AKT, which subsequently activates pro-survival signalling in injured cells; interestingly an increase in phospho-ERK1/2 was not observed [50]. Another in vivo study has shown that EVs derived from endothelial cells increase AKT phosphorylation in mouse cardiomyocytes [51].

The effect of EVs on angiogenesis was studied in a more complex assay with endothelial cells plated on a monolayer of fibroblasts. An increase in vessel area was observed in EV-treated cells under basal conditions following quantification, but this trend was not maintained in VEGF stimulated cells. However, an increase in vessel diameter was evident under basal conditions (Figure 5). It is possible in this more complex assay, that endothelial dependence on VEGF-mediated survival, via activation of AKT, is not as critical as in a collagen gel assay. The observation of enhanced vessel diameter following addition of EVs may reflect activation of alternative angiogenic pathways and potential transcriptional changes by EVs in this assay. Interestingly, human adipose mesenchymal stromal cell-derived EVs have been shown to stimulate neovascularisation and induction of growth factors such as angiopoetin/TIE2 and VEGF/VEGFR2 in a nude mouse model of fat grafting [52]. Fittingly, Angiopoetin-1/Tie2 signalling has been shown to regulate vessel diameter in vivo [53]. Hypoxia is well known to strongly promote angiogenesis signalling and to affect the angiogenic potential of EVs [54]. In our hands, the EVs obtained under hypoxia promoted a stronger, but not statistically significant, suppression of apoptosis compared to EVs isolated under normoxia (Figure 4). Exposure of adipose-derived MSCs to hypoxia has also been shown to affect the cargo and angiogenic capacity of their released EVs [52], suggesting that hypoxia-derived EVs may have greater angiogenic potential. The precise mechanism of EV-mediated enhancement of VEGF signalling observed remains obscure and could be due to effects of a biological cargo consisting of growth factors, RNA and miRNAs [55]. Future studies will be directed at understanding the stimulatory effects of EVs on AKT signalling in more detail.

Overall, our data defines a molecular mechanism for hUCMSC derived EVs on angiogenesis. Our investigations strongly suggest that the pro-angiogenic effects of hUCMSC-EVs are attributable to their capacity to inhibit apoptosis, potentially occurring via stimulation of VEGFR2-mediated AKT phosphorylation, therefore ultimately impacting endothelial cell survival during tubular morphogenesis. On the other hand, their impact on vessel morphology when studied in a co-culture setting, highlights the necessity to further study their biological impact in more complex models, as well as in appropriate in vivo models. Of interest, EVs from hypoxia preconditioned MSCs appear to have enhanced pro-angiogenic effects, and could potentially have greater therapeutic potency.

## Figures and Tables

**Figure 1 cells-11-03750-f001:**
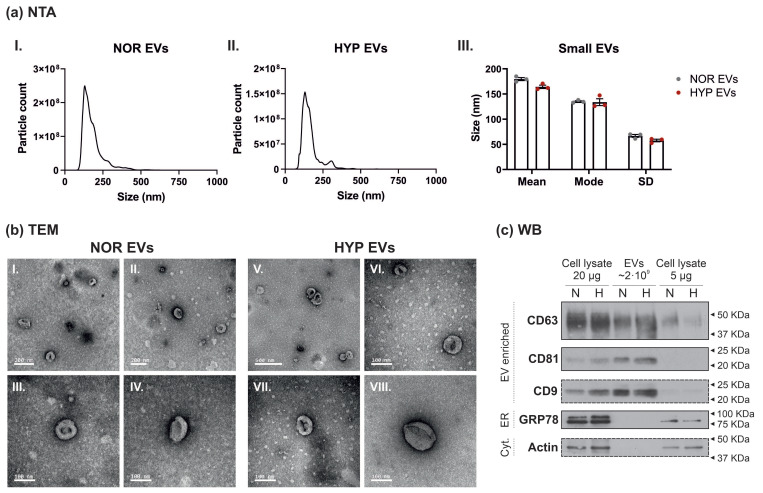
**Characterisation of EVs from NOR and HYP hUCMSCs.** (**a**) Nanoparticle Tracking Analysis (NTA) of small extracellular vesicles obtained from the conditioned medium of human umbilical cord-derived mesenchymal stromal cells (hUCMSCs) via differential ultracentrifugation following 24 h in culture under normoxic (NOR, (**aI.**)) or hypoxic (1% O2—HYP, (**aII.**)) conditions. (**aI.**,**aII.**) graphs show size distribution of the isolated EVs, and (**aIII.**) compares the three EV parameters obtained via NTA, showing no differences in size distribution among EVs from both culture conditions. Samples were measured in triplicate. (**b**) Transmission Electron Microscopy (TEM) images obtained from NOR/HYP hUCMSC-derived EVs confirming the presence of cup-shaped spheres within the size range expected for the EV population studied. Scale bars: I. and II. = 200 nm; III., IV., VI., VII., VIII. = 100 nm; and V. = 500 nm. (**c**) Western Blot (WB) confirming the presence of proteins widely considered to be enriched in small EVs (the 3 classical tetraspanins: CD63, CD81 and CD9) and the lack of other cell/organelle specific proteins (the cytoskeletal protein actin and the ER protein GRP78) demonstrating high enrichment of EVs following dUC isolation protocol. Dashed lines indicate proteins blotted after membrane stripping. N = normoxia, H = hypoxia, ER = endoplasmic reticulum, Cyt. = cytoskeleton.

**Figure 2 cells-11-03750-f002:**
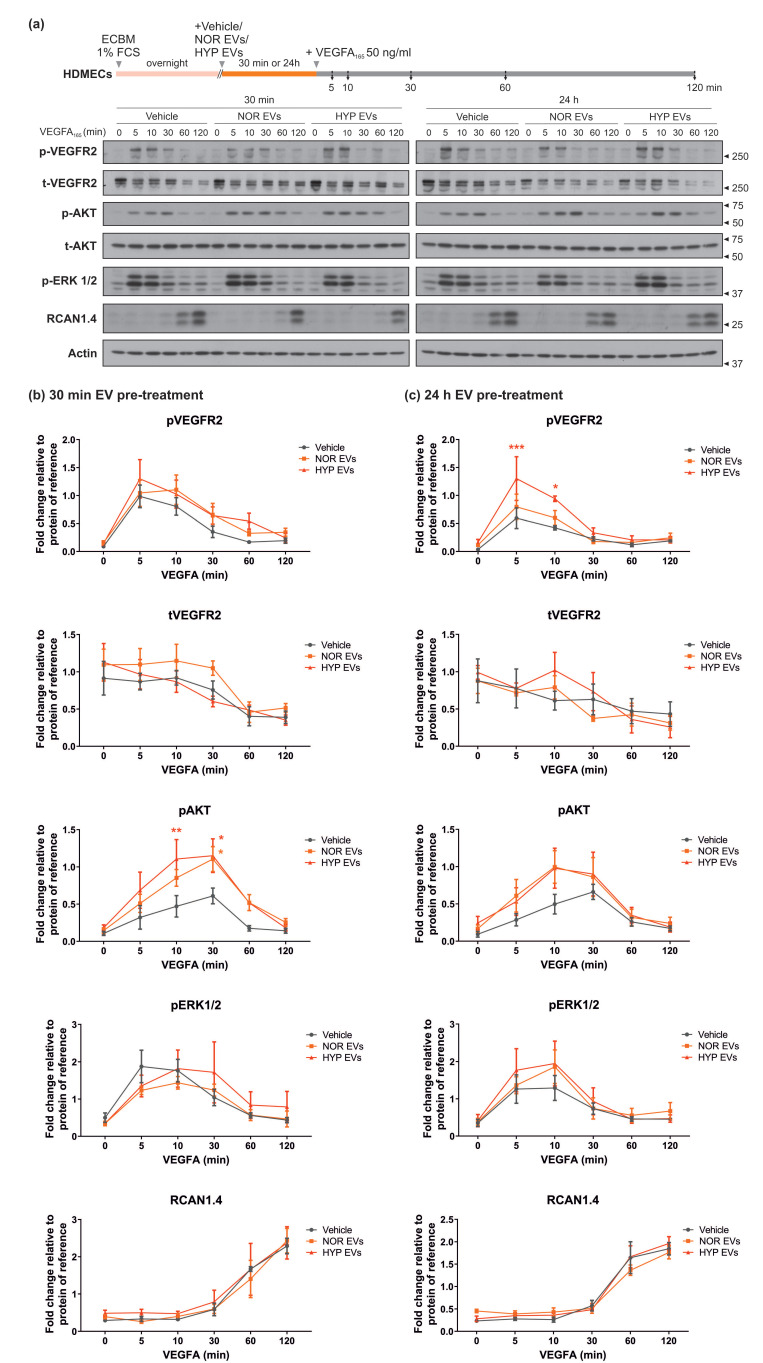
**hUCMSC-derived EVs alter VEGFA_165_ mediated VEGFR2 signalling.** (**a**) Human Dermal Microvascular Endothelial Cells (HDMECs) starved overnight with Endothelial Cell Basal Medium (ECBM) containing 1% Foetal Calf Serum (FCS) were pre-treated for either 30 min or 24 h with 2 × 10^7^ EVs from normoxic (NOR) or hypoxic (HYP) hUCMSCs. Following pre-treatment, cells were further stimulated with VEGFA_165_ (50 ng/mL) for different periods of time (5, 10, 30, 60 or 120 min). Cells were lysed and immunoblotted for phospho-VEGFR2 (Y1175), total VEGFR2, phospho-AKT (S473), total AKT, phospho-ERK1/2 (T202/Y204), RCAN1.4 and Actin. (**b**,**c**) show quantification graphs of pVEGFR2, VEGFR2, pAKT, pERK1/2 and RCAN1.4 over time after 30 min (**b**) or 24 h (**c**) EV pre-treatment. Data is normalised relative to the protein of reference. Graphs show mean ± SEM of n = 4 (**b**) or n = 3 (**c**) independent experiments. Statistical analysis: Two-way ANOVA followed by Tukey’s post hoc test, where * *p* < 0.05, ** *p* < 0.01 and *** *p* < 0.001.

**Figure 3 cells-11-03750-f003:**
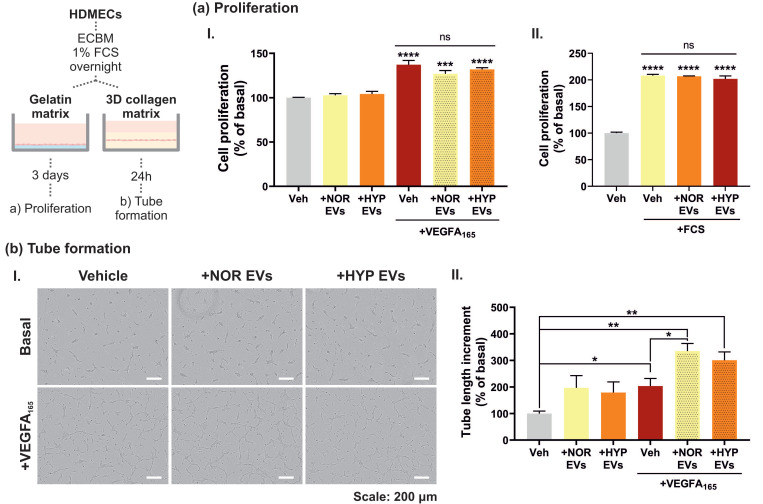
**hUCMSC-derived EVs impact tubular morphogenesis but not cell proliferation.** Serum starved HDMECs were either plated into gelatin-coated plates or between two layers of collagen I gel and stimulated with NOR/HYP EVs and/or VEGFA165 to assess cell proliferation or tube formation, respectively. (**a**) HDMECs plated over a gelatin matrix were pre-stimulated with NOR/HYP EVs for 30 min prior to further stimulation with VEGFA_165_ (50 ng/mL) (**aI.**) or Foetal Calf Serum (FCS, 10%) (**aII.**) for 3 days. Cells were then lysed with CellTiter-Glo™ and cell proliferation was measured via luminescence. Data is presented as mean ± SEM of percentage increase normalized to basal from n = 3 repeats. Statistical analysis: One-way ANOVA followed by Dunnett’s post hoc test, compared to basal conditions (vehicle, Veh), where ns (non-significant) *p* > 0.05, *** *p* < 0.001, and **** *p* < 0.0001; unpaired *t*-tests between groups showed no significant differences among +VEGFA_165_ or +FCS treated groups. (**b**) HDMECs plated over a 3D collagen matrix were seeded in the presence or absence of 10^7^ NOR/HYP EVs and later further stimulated with 10^7^ NOR/HYP EVs and/or 50 ng/mL VEGFA_165_ for 24 h by adding treatments 2× concentrated in the culture medium over the top collagen layer. Images were taken by the end of the experiment (**bI.**) representative image of n = 3 experiments). (**bII.**) Tube formation was measured using the AngiogenesisAnalyzer plugin in ImageJ. Data is presented as mean ± SEM of percentage increase normalized to basal from n = 3 independent experiments, each run in duplicate or triplicate. Statistical analysis: One-way ANOVA resulted in a *p* < 0.01; unpaired t-tests between groups are shown in the figure, where * *p* < 0.05, and ** *p* < 0.01.

**Figure 4 cells-11-03750-f004:**
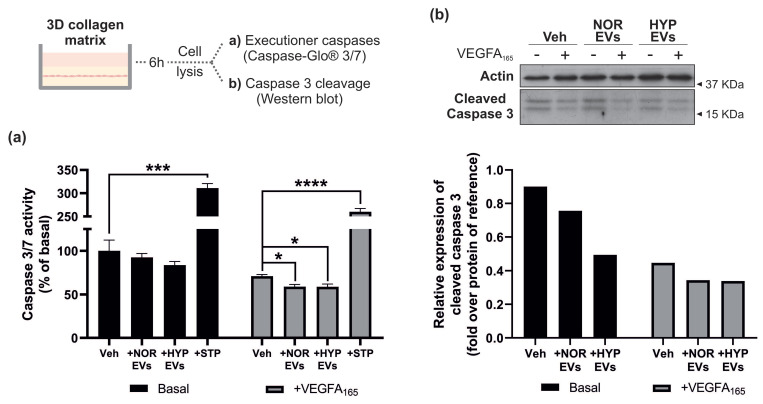
**hUCMSC-derived EVs reduce caspase-3/7 activation during tubular morphogenesis.** Serum starved HDMECs were plated between two layers of collagen I gel and stimulated with 10^7^ NOR/HYP EVs or 100 nM staurosporine (STP) and/or 50 ng/mL VEGFA_165_ for 6 h. Cells were lysed either with (**a**) Caspase-Glo^®^ 3/7 reagent and measured caspase-3/7 activity via luminescence, or (**b**) LDS buffer and immunoblotted with antibodies against cleaved caspase 3 and actin. Data in (**a**) is presented as mean ± SEM of percentage over the basal from n = 3 replicates, each measured in duplicate (mean of technical duplicates was used for analysis). Statistical analysis: two-way ANOVA showed *p* < 0.0001 for both studied factors: ±VEGFA and ±EV/STP treatments; unpaired *t*-tests between groups are shown in the figure, where * *p* < 0.05, *** *p* < 0.001 and **** *p* < 0.0001.

**Figure 5 cells-11-03750-f005:**
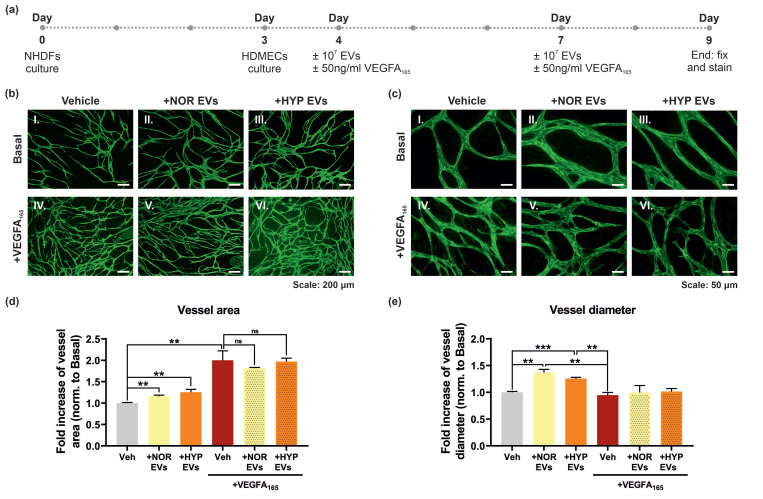
**hUCMSC-derived EVs impact vessel formation in a fibroblast-endothelial co-culture angiogenesis model.** (**a**) Normal Human Dermal Fibroblasts (NHDFs) were firstly plated as a feeding layer until confluent prior to the addition of HDMECs on day 3 in vitro. Cells were treated with 10^7^ NOR/HYP hUCMSC-derived EVs and/or VEGFA_165_ the day after adding HDMECs to the culture, and again 3 days after. Following 5 days of treatment, cells were fixed and immunostained with anti-VE-cadherin antibody and mounted. VE-cadherin fluorescence was detected to track vessel formation at both low (**b**) and high (**c**) magnifications (images from a representative experiment). (**d**) Total vessel area was quantified by AngioTool software. Data is presented as fold increase normalised to basal (data from 2 independent experiments, each ran in duplicate, n = 4, with 5–6 fields of view analysed per condition). (**e**) Average vessel diameter was quantified using REAVER analysis package. Data is presented as fold increase normalised to basal (data from 2 independent experiments, each ran in duplicate, n = 4, with 5–8 fields of view analysed per condition). Statistical analysis: One-way ANOVA resulted in *p* < 0.0001 (**d**) and *p* < 0.01 (**e**); unpaired *t*-tests between groups are shown in the figure, where ns (non-significant), *p* > 0.05, ** *p* < 0.01, and *** *p* < 0.001.

## Data Availability

Data is available upon request. Please contact corresponding authors: P.M and M.J.C.

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
