# Peer review of "Extracellular Vesicles from Human Umbilical Cord-Derived MSCs Affect Vessel Formation In Vitro and Promote VEGFR2-Mediated Cell Survival"

_cells, 2022, doi:10.3390/cells11233750_

Round 1

Reviewer 1 Report

In the article entitled “Extracellular vesicles from human umbilical cord-derived MSCs affect vessel formation in vitro and promote VEGFR2-mediated cell survival” by Ana Muniz-Garcia and colleagues, the authors show that:

1) EVs from human umbilical cord-derived MSCs can affect VEGFR2 signalling via altering the phosphorylation of AKT.

2) EV treatment results in inhibition of apoptosis, but not proliferation, in human microvascular endothelial cells.

3) Hypoxia-derived EVs appear to have a slightly enhanced effect.

4) EVs demonstrated a positive effect on vessel formation, evidenced by increased vessel networks with tubes of significantly larger diameters.

The work makes a good impression. The manuscript is high quality.

However I have some minor questions and comments.

1) Why did the authors use human umbilical cord-derived MSCs in their work? A number of studies point to that no VEGF is found in the conditioned medium of these cells. Meanwhile MSCs from other sources produce VEGF. Moreover, autologous adipose derived MSCs, for example, are easier to obtain in practice.

2) May be more information about VEGF stimulation in experiments is needed. Why is this stimulation of endothelium necessary?

3) «The final EV pellet was resuspended in PBS and either analysed or stored at -80°C until use. EVs were quantified using nanoparticle tracking analysis (NTA) with a Nanosight NS300 instrument (Malvern, UK) following 3 captions of 60 s each.» Did the Authors analyze the EVs after freezing in PBS? Which level of EV damage we can observe in this method of store?

4) Fig 2 (c). What is “H” on the graph?

Author Response

We thank the reviewer for their constructive suggestions and have addressed these below and in the revised manuscript.

1) Why did the authors use human umbilical cord-derived MSCs in their work? A number of studies point to that no VEGF is found in the conditioned medium of these cells. Meanwhile MSCs from other sources produce VEGF. Moreover, autologous adipose derived MSCs, for example, are easier to obtain in practice.

We are aware that MSCs from different sources may be able to affect VEGF signalling in a paracrine mode to different levels. In this study we focussed on the modulation of VEGF signalling caused by MSC-derived EVs, therefore, not necessarily caused directly by the secretion of VEGF itself (the real cause of the results here described still remains to be determined as stated in our discussion). Furthermore, we chose human umbilical cord MSCs due to their readily availability and younger donor age, therefore less susceptible to be affected by health status, lifestyle or the presence of comorbidities (i.e., adipose derived MSCs are often obtained from liposuction, meaning they are highly likely to be from obese individuals and potentially sourced from a pro-inflammatory environment). Because of this, we considered human umbilical cord may be a more consistent source of therapeutic EVs. We have added additional text in lines 397-399. “hUCMSCs are of particular interest, not only because of their ready availability, but also because the newborn status of the donors means the phenotype of the cells is less likely to be affected by their lifestyle or the presence of comorbidities.”

2) May be more information about VEGF stimulation in experiments is needed. Why is this stimulation of endothelium necessary?

In this study we wanted to investigate how EVs affect both basal and highly pro-angiogenic conditions. VEGF-A stimulation is known to robustly activate a number of intracellular signalling pathways and physiological responses in primary endothelial cells, allowing us to analyse pathways known to regulate cell proliferation and cell survival.

Section 3.2 lines 265-267, now reads “To assess whether the hUCMSC-derived EVs have any impact on endothelial cell signalling, we analysed the effect of the EVs on a number of VEGF-stimulated intracellular signalling pathways which regulate endothelial cell physiology.”

3) The final EV pellet was resuspended in PBS and either analysed or stored at -80°C until use. EVs were quantified using nanoparticle tracking analysis (NTA) with a Nanosight NS300 instrument (Malvern, UK) following 3 captions of 60 s each.» Did the Authors analyze the EVs after freezing in PBS? Which level of EV damage we can observe in this method of store?

EM imaging was performed after freezing in PBS and no damage was observed in the EV structure. All detected looked as those shown in the representative images: cup-shaped undamaged membranous structures. Following isolation, EVs were stored in small aliquots, meaning the freeze-thaw cycles were kept to a minimum upon use in the different studies here described. We have revised section 2.3 line 131-133 to now read: “The final EV pellet was resuspended in PBS and either analysed or stored at -80°C in small aliquots until use”

4) Fig 2 (c). What is “H” on the graph?

We have removed this from the figure.

Please note that the original Fig 6, the graphical summary, has now been moved to the relevant graphical abstract section online.

Reviewer 2 Report

In the manuscript by Gracia et al, “Extracellular vesicles from human umbilical cord-derived MSCs affect vessel formation in vitro and promote VEGFR2-mediated cell survival”. In this manuscript, authors have studied the effect of human umbilical cord derived MSCs EVs on angiogenesis. Manuscript is well designed and well executed with experiments and results. This manuscript will be interested to the scientific community working on EVs and regenerative medicine. I recommend addressing the following comments/suggestions before publications:

-How many passages were performed for hUCMSCs before isolation of EVs. As they may under differentiations after certain period or passage.

-Could you explain more in details about isolation of EVs method, the production yield of EVs is very less, how did you manage to perform all these experiments? Like how many EVs were isolated from how many cells?

-It is also recommended to study about the biological cargos responsible for this enhanced cell survival effect, maybe in discussion or future study? These maybe proteins or miRNAs.  

Author Response

We thank the reviewer for their constructive suggestions and have addressed these below and in the revised manuscript.

-How many passages were performed for hUCMSCs before isolation of EVs. As they may under differentiations after certain period or passage.

hUCMSCs were used maximum at P7, cryovials used for these were either P5 or P6, meaning cells underwent a maximum of 2 passages before EV isolation. We have now stated this in section 2.2 line 107-108.

-Could you explain more in details about isolation of EVs method, the production yield of EVs is very less, how did you manage to perform all these experiments? Like how many EVs were isolated from how many cells?

EVs were bulk isolated from the conditioned media of 15x10cm dishes per condition (normoxia or hypoxia) grown in parallel, therefore being at the same passage and from the same cryovial. The total EV yield from this number of plates was between 1.5x1010 to 2.5x1010 EVs (based on NTA count). Total cell number by the endpoint was not obtained for these experiments. The number of plates used was based on previous experience of culturing these cells and isolating EVs from them.

-It is also recommended to study about the biological cargos responsible for this enhanced cell survival effect, maybe in discussion or future study? These maybe proteins or miRNAs.

This is an interesting point and we have now added the following statement with supporting reference to our discussion on line 465-468. “The precise mechanism of EV-mediated enhancement of VEGF signalling observed remains obscure and could be due to effects of a biological cargo consisting of growth factors, RNA and miRNAs. Future studies will be directed at understanding the stimulatory effects of EVs on AKT signalling in more detail.”

Please note that the original Fig 6, the graphical summary, has now been moved to the relevant graphical abstract section online.

Reviewer 3 Report

In the manuscript entitled "Extracellular vesicles from human umbilical cord-derived MSCs affect vessel formation in vitro and promote VEGFR2 mediated cell survival," the authors investigated the angiogenic potential of extracellular vesicles derived from human umbilical cord-derived mesenchymal stromal cells, as well as their potential mechanism of action via the VEGF signaling pathway. The manuscript is very well-designed, with appropriate evaluations. This research is valuable in my opinion. Here are my comments and suggestions:

1. On page 2, lines 66-74, It would be helpful if the authors included and discussed some of the biomedical applications of MSC-derived EVs.

2. On page 3, subsection "2.2. Cell culture", the relative humidity should be reported as well.

3. In the "Materials and Methods" section, the number of replicates for each test should be specified.

4. The title of subsection 2.6 should be corrected.

5. On page 4, subsection "2.6. 3D collagen gel tube formation assay", It would be helpful if the authors briefly explained the gel preparation process.

6. The version of software used in this study, including ImageJ (line 179), ZenPro (line 218), AngioTool and REAVER (lines 220 and 221), should be specified.

7. On page 13, lines 395 and 396, "VEGF plays a critical role in … multiple signalling pathways." A reference is required for the statement. (For the reference: https://jnanobiotechnology.biomedcentral.com/articles/10.1186/s12951-020-00755-7)

8. Most of the references are not up-to-date. The authors should discuss the obtained results using the most recently published data.

Author Response

We thank the reviewer for their constructive suggestions and have addressed these below and in the revised manuscript.

  1. On page 2, lines 66-74, It would be helpful if the authors included and discussed some of the biomedical applications of MSC-derived EVs.

We have now expanded on our original text to include “Consequently, EVs have become of great interest in the field of regenerative medicine and have been reported to promote the repair of cardiac, liver, muscle, kidney and nervous tissue in vitro and in animal models” On page 2 line 70-72.

  1. On page 3, subsection "2.2. Cell culture", the relative humidity should be reported as well.

Our cell incubators do not have a relative humidity (RU) probe, considering that these are set at 37C, 5% CO2 and contain a full water tray we expect an estimated RU of 90-95%. We have now added this to the text in section 2.2 line 109 and line 112.

  1. In the "Materials and Methods" section, the number of replicates for each test should be specified.

We have included the number of replicates for each test in all figure legends.

  1. The title of subsection 2.6 should be corrected.

 This must have been a formatting issue, now corrected.

  1. On page 4, subsection "2.6. 3D collagen gel tube formation assay", It would be helpful if the authors briefly explained the gel preparation process.

We have now expanded on our original text in section 2.6 line 176-179. 3D collagen gel tube formation assay to include more detail on the gel preparation process. This section now reads: “Gels containing PureCol™ type I bovine collagen (Cellink, #5005-100ML) were prepared by mixing 10 × Ham's F-12 medium, 0.1 M NaOH, collagen type 1 (ratio 1:1:8), supplemented with bicarbonate solution (Invitrogen) to 0.117% v/v and Glutamax-I (Invitrogen) to 1 % v/v. The gels were allowed to set at 37 °C overnight in the incubator.” and “The top gel was allowed to set for two hours prior to cells being further stimulated with 50 ng/ml VEGFA165 and/or 107 EVs/well for 24h.”

  1. The version of software used in this study, including ImageJ (line 179), ZenPro (line 218), AngioTool and REAVER (lines 220 and 221), should be specified.

We have specified this for the pertinent programs in lines 224-229. REAVER is a code package run in Matlab, therefore it doesn’t have a version. We have also added more info on the microscope objectives used.

  1. On page 13, lines 395 and 396, "VEGF plays a critical role in … multiple signalling pathways." A reference is required for the statement.

We have now included 2 references to support this statement (line 406).

  1. Most of the references are not up-to-date. The authors should discuss the obtained results using the most recently published data.

We added a number of original references in the discussion to support our conclusions, especially around VEGFR2 and AKT activation. We have included some more recent references on review articles.

 Please note that the original Fig 6, the graphical summary, has now been moved to the relevant graphical abstract section online.